# Traditional Value Identity and Mental Health Correlation Among Chinese Adolescents

**DOI:** 10.3390/bs14111079

**Published:** 2024-11-11

**Authors:** Guofang Ren, Guanghui Yang, Junbo Chen, Qianru Xu

**Affiliations:** 1School of Education, Anyang Normal University, Anyang 455000, China; 2Faculty of Psychology, Beijing Normal University, Beijing 100875, China; 3Center for Machine Vision and Signal Analysis (CMVS), University of Oulu, 90570 Oulu, Finland

**Keywords:** adolescents, traditional values, mental health

## Abstract

This study explores the identity of traditional values among Chinese adolescents and its correlation with their mental health. A questionnaire survey, utilizing the Confucian Traditional Values Scale and the Secondary School Students’ Mental Health Scale, was conducted with 500 students from Grade 7 through the final year of college. Our results showed the following: (1) adolescents generally agree with traditional values; (2) there were no significant differences in overall agreement with traditional values among adolescents based on their gender, place of birth, class cadre status, only-child status, or academic major, though differences were observed in specific dimensions based on these variables; (3) traditional values identity varied across grade levels, with senior high school students showing notably higher identification than junior high school and college students, peaking in the second year of senior high school; and (4) there is a significant negative correlation between adolescents’ traditional values identity scores and their mental health scores, indicating that higher traditional values identity scores are associated with better mental health levels. These findings highlight the positive influence of traditional values on the development and well-being of Chinese adolescents, underscoring the importance of integrating these values into educational and developmental frameworks in China and other East Asian regions with similar cultural backgrounds.

## 1. Introduction

In the 21st century, the rapid pace of technological advancements has led to changes in daily life that are far more substantial than those experienced by our ancestors. For countries with a rich history and traditional culture, such as China, traditional values play an essential role in shaping the moral framework of both individuals and society. Within Chinese communities, traditional values represent the inertial standards of evaluation and perspectives regarding the significance and importance of objective entities, such as people, events, and objects. These traditional values, developed over an extensive period in the nation’s history and predominantly based on Confucianism, have been the dominant social values in Chinese society for thousands of years and rooted in many other East Asian countries that share similar cultural influences [1,2].

Much like many other civilizations [3], newly industrialized East Asian countries such as China have experienced a profound impact from “modern” values, particularly on the younger generation growing up in this new environment. Influenced by modern life and the values inherited from their ancestors, the perceptions and acceptance of traditional values by younger generations could have significant implications for both them and society. Specifically, values like respect for elders [4], honesty, and humility [5] are central to cultural heritage and help shape personal values system, forming the cornerstone of social stability and harmony. Interestingly, while there has been considerable interest and research on “traditional China” and “Chinese values” in Western societies (see a brief review in [6]), research within China has mainly focused on specific groups, such as college students, rather than examining a broader developmental range across adolescence. Consequently, there remains a relative lack of studies that take a more diverse developmental approach to understanding traditional values.

### 1.1. Studies on Traditional Values

Compared to its rich history, studies on Chinese traditional values have been relatively limited and span a shorter historical period. For instance, in 1995, ref. [7] conducted a cross-cultural comparative study examining the dominant values in China and the United States, confirming that Chinese values are influenced by Confucian ideals, while American values are shaped by Judeo–Christian principles. The study observed that Chinese participants tended to prioritize values such as respect for authority, humility, obedience, self-restraint, and diligence, whereas American participants placed greater importance on religious beliefs, honesty, and helpfulness. However, these findings reflect general trends rather than fixed characteristics and should be interpreted within a broader cultural context, taking into account multiple factors, such as historical and socioeconomic conditions. Separately, ref. [8] highlights the significant role of Confucian principles in shaping the lives and worldviews of the Chinese people, both historically and in contemporary society and developed the “Chinese Values Scale”, grounded in Confucianism.

In more recent years, traditional values have regained attention in Chinese society, and it has been observed that traditional values continue to hold an important position in the value system of college students. For example, ref. [9] found that the traditional values among college students exhibit diversity, variation, and contradiction. The majority of college students agreed with the positive aspects of traditional values and disagreed with most negative aspects. Further, one study explored the influence of university environments on the identification of traditional values, and the results showed that while college students’ attitudes towards traditional values are subject to change under university influence, they gradually stabilize [10]. Additionally, ref. [11] found that college students accept and agree with traditional values to a significant extent.

To summarize, previous studies have predominantly focused on developing tools to measure values and investigating traditional values within specific groups, notably college students or general adults [9,10,11]. However, there is a lack of research addressing the complete developmental formation of values across the entire adolescence period. In this sense, this paper aims to bridge this gap by examining a broader range of students (i.e., from junior high and high school through to the college) to explore potential differences in the identification of traditional values across various stages of adolescent development.

### 1.2. Research Related to Traditional Values and Mental Health

While the concept of “mental health” in its modern clinical sense is often viewed as a Western construct that has only recently gained traction in the Chinese community, the pursuit of mental and emotional well-being has been deeply rooted in Chinese culture for centuries, shaped by traditional philosophies like Confucianism, Taoism, and Buddhism [12]. For instance, Buddhism encourages individuals to understand concepts like non-self, karma (cause and effect), interconnectedness, and impermanence as a way to address suffering and focus on self-improvement as part of a broader whole. Taoism promotes finding harmony by aligning with the Law of Nature, embracing contradictions, and accepting oneself and life events through the principle of “Wu-Wei” (inaction). Confucianism, in turn, encourages self-cultivation and human-heartedness, emphasizing social harmony, relational hierarchy, and adherence to social norms as ways to maintain order and fulfill communal roles [13,14]. According to the literature [15,16], traditional values such as emphasis on the family and balance can provide guidelines for one’s behavior and emotional responses in public. These guidelines help individuals maintain inner balance and a stable sense of self-identity when facing life’s challenges by providing culturally grounded responses that promote socially acceptable behavior while meeting their own emotional needs. At the same time, these influences may also present challenges that could lead to negative effects. Consequently, the traditional values of East Asian societies, such as China, may shape mental health concepts and coping strategies in ways that differ significantly from those commonly seen in Western European societies.

In sum, investigating the relationship between one’s identification with traditional values and its correlation with mental health is an intriguing area of study. Studies with college students have shown that Confucian and Taoist values significantly moderate the relationship between life events and mental health, and have a positive impact on mental health [17]. However, there is a lack of in-depth investigation into how adolescents’ identification with traditional values evolves through different stages of schooling, as well as a need for targeted interventions to further promote mental health and enhance cultural self-confidence. In this regard, we intend to explore the relationship between adolescents’ traditional values and mental health. We hypothesize the following: (1) the degree of identification with traditional values varies among different adolescent groups, and (2) adolescents’ identification with traditional values is correlated with their level of mental health.

## 2. Materials and Methods

### 2.1. Participants

This study employed a convenience sampling method, targeting students from freshmen to seniors at Anyang Normal University (for college students) and Yuhua Experimental School (for secondary school students), both located in Henan Province, Central China. A total of 500 questionnaires were distributed, with 489 valid responses collected, resulting in a validity rate of 97.8%. The sample included 146 junior high school students (29.9%), 139 high school students (28.4%), and 204 college students (41.7%). Among the respondents, there were 182 male students (37.2%) and 307 female students (62.8%), with 130 urban students (26.6%) and 359 rural students (73.4%). There were only 115 children (23.5%) and 66 class cadres (13.5%). Among the high school and college students, 195 were studying liberal arts (56.9%), and 148 were studying sciences (43.1%). The procedures of the study complied with the Declaration of Helsinki and were approved by the ethical committee of Anyang Normal University. All participants provided written informed consent prior to participation.

### 2.2. Measurements

#### 2.2.1. Traditional Values Scale

To measure the traditional value identity of students, we utilized the “Confucian Traditional Values Scale” compiled by [18]. This scale comprises 40 value items, evaluated on a 4-point scale, where “not important at all” was scored as 1, “somewhat important” as 2, “quite important” as 3, and “very important” as 4. The scale encompassed five dimensions: familialism, humility and respect, hard work, unity and harmony, as well as face-saving relations. The reliability scores for these five dimensions are 0.87, 0.82, 0.60, 0.84, and 0.71, respectively, ranging from 0.60 to 0.87, indicating that the scale demonstrates overall good reliability [18].

#### 2.2.2. Mental Health Scale for Secondary School Students

To assess mental health, we employed the “Secondary School Students” Mental Health Scale” developed by [19], which comprises 60 items and employs a 5-point scoring system. Scores were assigned as follows: “none” scores 1 point, “mild” scores 2 points, “moderate” scores 3 points, “severe” scores 4 points, and “very severe” scores 5 points. We adopted a total score assessment method, using the mean average score of the mental health scale to evaluate the mental health status of secondary school students. Scores ranging from 1 to 1.99 indicate no mental health problems; 2 to 2.99 suggest mild mental health issues; 3 to 3.99 denote moderate mental health issues; 4 to 4.99 imply more serious mental health problems; and a score of 5 indicates very serious mental health issues. The correlation coefficients between the 60 items of the scale and its total score ranged from 0.4 to 0.76, demonstrating good item discrimination. The test–retest reliability of the 10 subscales ranged from 0.72 to 0.91, with internal consistency reliabilities between 0.60 and 0.86, and split-half reliabilities between 0.63 and 0.87. The correlation coefficients between the subscales and the total scale range from 0.77 to 0.87, indicating that the scale has good structural validity [19].

#### 2.2.3. Statistical Analysis of Data

SPSS 22.0 statistical software was used for statistical analysis.

## 3. Results

### 3.1. Overall Situation of Adolescents’ Identification with Traditional Values

Table 1 shows the details of traditional values at both the overall level and in each dimension. It indicates that adolescents have a high overall level of identification with traditional values. The highest scores are observed in unity and harmony, followed by familialism, with the lowest scores in face-saving relationships. The results of repeated measures of ANOVA reveal a significant main effect of traditional values (*F*(4,485) = 206.84, *p* < 0.001). Subsequent post-hoc multiple comparison analysis shows differences in traditional values across each dimension among adolescents. Specifically, familialism scored significantly higher than humility and respect, face-saving relations, and hard work (all *p*-values < 0.001). Humility and respect scored significantly higher than face-saving relations and hard work (all *p*-values < 0.001). In addition, unity and harmony scored significantly higher than humility and respect, face-saving relations, and hard work, with all *p*-values < 0.001.

### 3.2. Demographic Group-Based Comparison of Adolescents’ Identification with Traditional Values

Adolescents were further grouped based on their demographic variables to compare differences in traditional values across these groups. The analysis revealed no gender differences in the overall identification with traditional values among adolescents. However, significant gender differences were found in the identification with humility and respect (t = 3.61, *p* < 0.001, Cohen’s *d* = 0.306), face-saving relationship (t = 3.07, *p* < 0.01, Cohen’s *d* = 0.305), and hard work (t = 3.60, *p* < 0.001, Cohen’s *d* = 0.348), with male students scoring significantly higher than female students.

There was no significant difference between urban and rural adolescents in the overall score and across each dimension of identification with traditional values. However, adolescents who were class cadres scored higher than their peers on the face-saving relations dimension (t = 3.07, *p* < 0.01, Cohen’s *d* = 0.305). Non-only children showed higher scores in face-saving relations compared to only children in their families (t = 2.23, *p* < 0.05, Cohen’s *d* = 0.238). There was also a significant difference in familialism identification based on academic major, with liberal arts students exhibiting stronger familistic tendencies compared to those studying science (t = 2.81, *p* < 0.01 Cohen’s *d* = 0.306).

### 3.3. Trends in the Identification of Traditional Values Among Adolescents by Grade Level

Figure 1 illustrates the trends in overall identification with traditional values among adolescents across different grade levels.

There were significant grade-level differences in overall traditional values identification (*F*(9,479) = 2.02, *p* < 0.05, *η_p_*^2^ = 0.037). As depicted in Figure 1, the identification with traditional values among grade 11 high school students (senior two) was significantly higher compared to grade 7 (junior one) and grade 9 students (junior three), as well as second- to fourth-year college students (all *p*-values < 0.05). The traditional values identification of grade 9 students (junior three) was significantly lower than that observed in grade 8 (junior two), grade 10 (senior one), and grade 11 (senior two) students, as well as first-year college students (all *p*-values < 0.05).

### 3.4. Variations in Adolescents’ Identification with Traditional Values Across Educational Stages

Upon analyzing the level of identification with traditional values across different educational stages, significant differences were observed both overall and within each dimension of traditional values identification among junior high school, high school, and college students. Table 2 details these findings.

### 3.5. Correlation Analysis Between Adolescents’ Identification with Traditional Values and Mental Health Levels

First, the overall mental health status among adolescents was assessed. Out of 489 students, 390 exhibited no mental health issues (scores ranging from 1 to 1.99), 86 experienced mild mental health problems (scores from 2 to 2.99), 13 faced moderate mental health problems (scores from 3 to 3.99), and none were found to have severe or very severe mental health problems (scores 4 to 5).

Subsequently, a Pearson correlation analysis was conducted to examine the relationship among different dimensions of traditional values identification and adolescents’ mental health level. The results of this analysis are presented in Table 3. There is a significant negative correlation between adolescents’ overall identification with traditional values and their mental health scores, as well as between their mental health scores and all assessed dimensions of traditional values, including familialism, humility and respect, face-saving relations, unity and harmony, and hard work. These results suggest that higher scores on traditional values are associated with lower mental health scores. Notably, within this context, lower mental health scores indicate better mental health levels.

Finally, a stepwise regression analysis was conducted to explore the predictive relationship between adolescents’ identification with traditional values (overall traditional values score and five dimensions including familialism, humility and respect, face-saving relations, unity and harmony, and hard work) and their mental health levels. In this model, adolescents’ mental health level served as the dependent variable, while the aforementioned traditional values served as predictor variables. The results indicated that out of all the predictors, only the overall identification with traditional values emerged as a significant predictor in the regression model, accounting for 9.7% of the variance in mental health levels (see Table 4 for more details). This suggests that a general alignment with traditional values is a substantial predictor of adolescents’ mental health status.

## 4. Discussion and Conclusions

As evidenced in Table 1, the overall average score of adolescents’ identification with traditional values is 3.13, surpassing the theoretical mean of 2.5. This indicates a high degree of identification with traditional values among adolescents, aligning with the findings of [20]. However, as Table 1 shows, there are some variations among adolescents across different dimensions of traditional values. Specifically, the score for familialism is significantly higher than those for humility and respect, face-saving relationships, and hard work. The score for humility and respect is significantly higher than those for face-saving relations and hard work, but lower than the score for unity and harmony. The score for face-saving relations is lower than those for unity and harmony and hard work, whereas unity and harmony score significantly higher than hard work. These results suggest that contemporary Chinese adolescents remain deeply influenced by Chinese traditional values, particularly in terms of filial devotion, and give priority to family values. This is evident in their higher scores in familialism, as compared to other dimensions such as humility and respect, face-saving relations, and hard work. This pattern also aligns with the findings from a decade ago, which revealed that university students strongly endorse filial obligations towards the elderly [21]. This sustained prominence of familial values underscores the lasting impact of traditional Chinese cultural norms on the value system of today’s youth. In the rapidly changing society, adolescents’ strong endorsement of familialism may also be seen as a reflection of their need for security and stability. A supportive family environment characterized by close interactions and respect for ethical and social principles fosters a sense of stability and belonging, making it a primary support mechanism [22]. In the Chinese context, adherence to familialism can provide essential emotional and social support, which is crucial for shaping adolescents’ social health and values and could, therefore, be further emphasized in future educational frameworks. Moreover, the observed emphasis on humility and respect, as well as unity and harmony, can be attributed to the quality of education adolescents receive. The education not only encourages them to think about others and conform to social norms and school disciplines, but also emphasizes harmony, a core value in Chinese culture, which serves as a foundation for many other traditional values [23]. However, influenced by contemporary ideologies, adolescents are less reliant and obedient to authority than in the past, leading to notable changes in their attitudes towards authority, especially among urban adolescents [24]. This shift may account for their relatively lower scores in face-saving relations. In addition, their engagement with diverse peer groups in educational environments is likely to enrich their comprehension of unity and harmony’s significance, as peer contexts play an important role in their academic success and social adaptability [25].

There is no significant difference in the overall identification with traditional values among adolescents based on gender. However, significant differences were observed in specific dimensions such as humility and respect, face-saving relations, and hard work, with male students consistently scoring higher than their female counterparts. This discrepancy may stem from the societal pressure exerted on males to conform to traditional masculine roles. From an early age, boys are often taught that “boys must be brave”, reinforcing the belief that they should overcome difficulties and work hard. Furthermore, as highlighted by a recent study [26], the impact of traditional gender role attitudes on income disparities between males and females in contemporary Chinese society shows that men’s gender role attitudes tend to be more traditional than those of women. This disparity may be interpreted as a form of societal stress on males to uphold these roles in aforementioned dimensions. In terms of urban and rural comparisons, no significant differences were observed in the overall scores and across each dimension of traditional values identification. This lack of disparity might be attributed to the current era of information technology, where both urban and rural areas have largely benefited from comprehensive network coverage. Consequently, adolescents in both settings share a common backdrop of receiving information and concepts of the new era. These findings are consistent with a meta-analysis indicating that, although gender and spatial (rural vs. urban) inequalities still exist, they have been narrowing over time [27]. Future research should aim to cover a broader range of samples to further validate these findings.

There is no significant difference in the general identification with traditional values among adolescents based on their roles in class (whether they are class cadres or not), their status as only children, or their subject specialties. However, our study revealed higher scores in face-saving relations among adolescents who are class cadres and those who are not the only children in their families. This disparity may arise from the influence of the social praise effect, as class cadres and non-only children may seek approval and recognition from others, aligning more closely with traditional values in face-saving. Additionally, there is a notable difference in identification with familialism in terms of the subject specialties, with liberal arts students showing higher levels of identification of familialism compared to science students. This trend may be linked to their academic backgrounds, as liberal arts students are more exposed to traditional culture through in-depth studies of ancient literature and history. Furthermore, recent research [28] suggests that students in liberal arts are often influenced by their family background and parental values when choosing their major. This familial bond may also partly explain the heightened identification with familialism observed in these students.

Adolescents’ identification with traditional values shows variability across different grade levels and educational stages, both generally and within specific dimensions (refer to Figure 1 and Table 2 for detailed results). Specifically, adolescents’ identification with traditional values is markedly lower at the junior high school level, particularly in the third year (grade 9), then heightens in high school (especially in grade 11). Upon entering college, this identification initially decreases and then progressively stabilizes at the university level. The trend among college students aligns with previous research, which indicates that college students’ attitudes toward traditional values are, in general, positive and stabilize over time [9,10,11]. Our study, however, is the first to examine the variation of Chinese traditional values across the stages of adolescence, including samples from junior high schools to college students. This represents a novel finding in its own right. Generally, adolescents’ understanding of the world and themselves develops alongside their educational age. The observed variation in culturally related values across different school ages is also consistent with results obtained from other cultures (e.g., [29]). In addition, teenagers around grade 9 (ages 13–15) have shown statistical trends of sudden increases in risk-taking behavior [30,31], and may start to challenge authority more readily, leading to a decreased emphasis on traditional values. However, as adolescents mature and progress into senior high school, they demonstrate enhanced perspective-taking abilities compared to their younger counterparts and begin to develop a more profound understanding of the world [31]. This cognitive and emotional development may facilitate a deeper appreciation and cherishing of their traditional values identity. At the university stage, although students’ interactions become more extensive, they have already developed relatively good social skills in considering others’ perspectives [31]. This development may account for the changes observed during the freshman year of college and the subsequent stabilization of their identification with traditional values.

Additional correlation and regression analyses were conducted to examine the relationship between adolescents’ identification with traditional values, both in general and across various dimensions, and their mental health levels. As indicated in Table 3, traditional values such as familialism, humility and respect, face-saving relations, unity and harmony, hard work, and overall traditional values all demonstrate significant negative correlations with mental health levels. This result supports our hypothesis and suggests that a higher degree of identification with traditional values corresponds to lower mental health scores, implying better overall mental health. Moreover, adolescents’ overall identification with traditional values accounted for 9.7% of the variance in their mental health levels. This implies that a general alignment with traditional values is a predictor of mental health, indicating that traditional values play a crucial role in fostering psychological health among adolescents, particularly within the East Asian cultural context. This relationship could be attributed to the stability and sense of identity that traditional values provide within Chinese community, which may be especially significant during the adolescent years. These results are also consistent with previous studies in college students, which have demonstrated that Confucian and Taoist values positively influence mental health [17].

It is worth noting that in our sample, nearly 80% (390 out of 489) of students reported no mental health issues, with no cases classified as severe or very severe. This finding contrasts with international data, which typically report higher rates of mental health challenges among adolescents. For instance, it has been estimated that 49.5% of U.S. adolescents have experienced a mental health disorder at some point, with over 22% of cases involving severe impairment [32]. This discrepancy may suggest either a lower prevalence of mental health issues in this sample or cultural influences on how mental health is perceived and reported. Although modern understandings of mental health are increasingly recognized, mental health remains a relatively new concept in Chinese and broader East Asian societies, where stigma surrounding mental illness is still pronounced [33,34]. Research suggests that individuals of East Asian heritage often encounter greater stigma surrounding mental illness compared to Western norms and may show reluctance in disclosing mental health concerns, even to close friends and family (for a review, see [13]). These cultural influences are essential to consider when interpreting self-reported mental health data and developing relevant educational and treatment strategies tailored to adolescent mental health in East Asian populations.

As previously mentioned, although the concept of “mental health”, with its Western origins, has only recently been introduced in Chinese society, traditional Chinese philosophies have long offered unique perspectives on mental well-being. For instance, Taoism emphasizes harmony with nature, “Wu-Wei” (inaction), and an acceptance of life’s natural rhythms as paths to inner peace. Instead of promoting active control over one’s circumstances, Taoism encourages adaptability and alignment with the natural flow, fostering emotional resilience through acceptance and prioritizing self-transcendence [12]. Buddhism, akin to Taoism, encourages for a non-judgmental awareness and acceptance of the present moment, primarily achieved through meditation. This practice allows individuals to distance themselves from suffering, facilitating a journey toward liberation and enlightenment [14,35]. Confucianism, on the other hand, emphasizes “Zhong He” (balanced harmony), which encourages introspection and self-examination to address mental distress caused by narrow-mindedness or an overemphasis on unimportant matters [36]. Despite their distinct metaphysical beliefs, Taoism, Buddhism, and Confucianism each uniquely shape daily life in East Asia [14]. While these traditional Chinese philosophies may not address all mental health issues and may have contributed to a general hesitancy within Chinese and broader East Asian cultures to seek psychological help, they offer culturally resonant methods for processing defensive emotions and managing aggression. Integrating these philosophies into modern therapeutic practices could prove beneficial, particularly within an East Asian context. In this paper, we focus on exploring adolescents’ identification with traditional values, specifically within the framework of Confucianism, as it plays a central role in shaping Chinese cultural norms. This focus is particularly relevant to our sample from Central China, a traditional region of Han cultural development where Confucianism may exert an even stronger influence [37]. Future research might consider further exploring adolescents’ identification with other traditional values and examining how these values relate to mental health. Incorporating this understanding into adolescent mental health education could help youth better express and manage their mental health concerns, while also supporting the development of more tailored therapeutic strategies that integrate ancient philosophical wisdom.

In sum, our study revealed the following: (1) Adolescents from junior high school to college generally acknowledge traditional values. (2) An overall identification with traditional values does not significantly vary based on gender, place of origin, class cadre status, only-child status, or academic major. However, significant differences were observed in specific dimensions such as familialism and face-saving relations. (3) Significant variations in traditional values identification were observed across grade levels and educational stages, with the lowest level of identification in junior high school (grade 9) and a peak in high school (grade 11). At the college age, their identification with traditional values tends to stabilize. (4) A significant correlation was found between adolescents’ mental health levels and their identification with traditional values, both generally and across various dimensions. Overall, adherence to traditional values positively contributes to adolescents’ mental health status.

In conclusion, this study shows that contemporary young adolescents in China generally appreciate traditional cultural values, which significantly and positively influence their mental health. In this sense, there is a crucial need for schools to stress these values, particularly among junior and senior high school students, to strengthen their connection with these traditions and highlight aspects beneficial to their mental well-being. Future research should not only expand in terms of sample size and diversity but also delve into the structural and developmental aspects of traditional values among adolescents. Such research is important in enhancing our understanding of the role traditional values play in shaping individual mental health. This insight could be particularly valuable in East Asia, which shares similar cultural roots and many of these traditional values, potentially aiding in improving the mental health of adolescents in this region in general.

## Figures and Tables

**Figure 1 behavsci-14-01079-f001:**
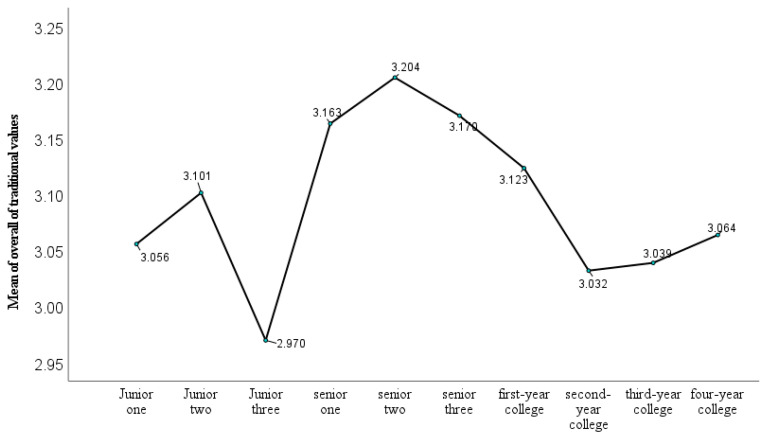
Trends in traditional values identification among adolescent by grade level. Junior one to senior three denotes grades 7 to 12.

**Table 1 behavsci-14-01079-t001:** Descriptive statistics and multiple comparison.

Values Dimensions	1 Familialism	2 Humility and Respect	3 Face-Saving Relations	4 Unity and Harmony	5 Hard Work	6 Overall
M ± SD	3.39 ± 0.42	3.00 ± 0.48	2.80 ± 0.49	3.40 ± 0.46	2.85 ± 0.61	3.13 ± 0.35
Comparisons		1 > 2, 1 > 3, 1 > 5, 2 > 3, 2 < 4, 2 > 5, 3 < 4, 4 > 5 (all *p*-values < 0.001)				

Note: 1 = familialism, 2 = humility and respect, 3 = face-saving relations, 4 = unity and harmony, 5 = hard work.

**Table 2 behavsci-14-01079-t002:** Comparison of differences in traditional values identification across educational stages.

Variables	Junior School (*N* = 146)	High School (*N* = 139)	College (*N* = 204)	
	M ± SD	M ± SD	M ± SD	*F*
Familialism	3.25 ± 0.351	3.36 ± 0.461	3.51 ± 0.404	16.306 ***
Humility and respect	3.04 ± 0.381	3.18 ± 0.418	2.84 ± 0.538	24.305 ***
Face-saving relations	2.91 ± 0.421	2.98 ± 0.450	2.62 ± 0.489	31.513 ***
Unity and harmony	3.30 ± 0.424	3.38 ± 0.474	3.48 ± 0.463	6.721 **
Hard work	2.71 ± 0.578	2.98 ± 0.597	2.85 ± 0.620	7.366 **
Overall	3.10 ± 0.277	3.21 ± 0.351	3.10 ± 0.389	5.209 *

Note: * *p* < 0.05, ** *p* < 0.01, *** *p* < 0.001.

**Table 3 behavsci-14-01079-t003:** Correlation between traditional values identification and mental health levels.

	Familialism	Humility and Respect	Face-Saving Relations	Unity and Harmony	Hard Work	Overall
Mental health	−0.291 **	−0.238 **	−0.188 **	−0.264 **	−0.110 *	−0.215 **

Note: * *p* < 0.05, ** *p* < 0.01.

**Table 4 behavsci-14-01079-t004:** Regression analysis results.

Output Variable	Predictor Variable	B-Value	*t*	R^2^	F
Mental health	Overall traditional values	−0.449	−7.324 ***	0.097	53.637 ***

Note: *** *p* < 0.001.

## Data Availability

The data used and analyzed during the current study are available from the corresponding author on reasonable request.

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
