# Peer review of "Traditional Value Identity and Mental Health Correlation Among Chinese Adolescents"

_behavsci, 2024, doi:10.3390/bs14111079_

Round 1

Reviewer 1 Report

Comments and Suggestions for Authors

Overall, this is a good article and is already pretty much polished. In my view, only a few editing should be done, like:

Line 35: Confucianism is an important source of values, but Taoism is too. Maybe you could be more precise about the difference in treatment and influences of values between Confucianism and Taoism. (This is an overall remark: why somehow are Taoist values discarded? When, in fact, they are essential for mental health and are even sometimes considered as what prevented Chinese tradition from developing its therapeutic methods.)

line 71 "measuring",

line 90-91: "strategies" (here, the sentence should be checked)

line 152: "Familialism"

line 236: " of...?"

line 325: "Familialism"

line 335: "Chiese"

(I am sure there are other typos). 

The discussion could go a bit further regarding the reasons for the general endorsement of traditional values: could this be related to a general atmosphere where a need for safety and security makes teenagers see familialism as an essential value?

These few remarks do not cast a shadow on the quality and interest of the article. 

Author Response

Thank you for your valuable suggestions, which have greatly contributed to refining our study. In the revised manuscript, you’ll find a clean version with key changes highlighted in yellow, followed by the original version with tracked changes for more detailed revisions. A detailed response is also provided in the attached document. 

Reviewer 2 Report

Comments and Suggestions for Authors

The research paper offers a valuable contribution by examining the correlation between adolescents' identification with traditional values and mental health in China. The study utilizes robust measurement tools and a considerable sample size to yield insightful results. However, several areas could benefit from further clarification and revision.

Please add in the headline and/or the abstract, the research national context (China). Eventhough it is mentioned in the text of the paper, for other scholars it might be a helpful information within the abstract or headline to identify its national context of your study. Furthermore I want to highly recommend to provide more specific regional informations of where the schools are located in China, that could be more helpful for international scholars to have a introduction into its local characteristcs.

The claim on page 2 (46-48) that "despite considerable interest and research in 'traditional China' and 'Chinese Values' in Western societies, the extent of related research within China remains relatively limited" could be misleading and should be rephrased. The literature does in fact show considerable research on traditional values within China, though the focus has often been on specific groups, such as college students, rather than a broader developmental range across adolescence. It might be more accurate to state that the diversity and developmental scope of studies on traditional values are limited, rather than implying a lack of research altogether.

Regarding the representation of earlier studies of such of Shen (1995), I want to mention, that the study indeed found differences between Chinese and American values, with Confucianism influencing Chinese values and Christianity influencing American values. However, the description of traits such as “humble, hardworking, and respectful of authority” vs. “religious, honest, and helpful” might oversimplify Shen's findings. I suggest therefore that it is essential to provide nuanced interpretations rather than generalized traits to avoid reinforcing cultural stereotypes.

The statement that “mental health is a Western concept that was only recently introduced to the Chinese community” needs more clarification when it comes to an international audience. While Western psychological frameworks have influenced modern Chinese mental health discourse, traditional Chinese medicine and philosophies such as Confucianism, Daoism, and Buddhism have long addressed mental well-being, albeit in different terms. Thus, while "mental health" in its modern clinical sense may be newer, concepts of mental and emotional balance have deep roots in Chinese culture. In this context it is noteworthy that 390 out of 489 students reported no mental health issues (page 6 line 205), which is remarkably high compared to international averages. This finding warrants a discussion on whether it reflects a cultural bias in self-reporting or a lack of social awareness about mental health. The authors could explore if the framing of mental health as a “Western concept” might contribute to underreporting or different interpretations of mental health symptoms.

I might be worthy to explore whether the high percentage of students without mental health problems correlates with cultural factors influencing self-reporting? Furthermore, would it be possible to include a discussion on traditional Chinese approaches to mental health, contrasting these with Western models? The results concerning specific categories such as gender differences are correctly represented in the tables, where male students scored higher in certain dimensions such as humility, face-saving relations, and hard work. The findings seem to be consistent with the statistical tests reported, including t-tests and Cohen's d values that indicate significant differences.

Finally yet importantly, the language throughout the paper could benefit from refinement. Specifically: On page 2, lines 52 ("and and") and 61 ("an an") need correction; on page 5, lines 189-190 appear fragmented by accident. A thorough proofreading to address these and other stylistic inconsistencies would enhance readability of your paper.

Overall, the study presents interesting findings but would benefit from refining language, addressing cultural context more deeply, and ensuring nuanced interpretation of previous research.

Comments on the Quality of English Language

see above

Author Response

(The authors gave the same response as above.)
